# Fermented Kamut Sprout Extract Decreases Cell Cytotoxicity and Increases the Anti-Oxidant and Anti-Inflammation Effect

**DOI:** 10.3390/foods12112107

**Published:** 2023-05-24

**Authors:** Hosam Ki, Jun-Seok Baek, Hye-Jin Kim Hawkes, Young Soo Kim, Kwang Yeon Hwang

**Affiliations:** 1Materials Science Research Institute, LABIO Co., Ltd., Seoul 08501, Republic of Korea; msri07@labio.kr (H.K.); msri08@labio.kr (J.-S.B.); 2Department of Biotechnology, Korea University, Seoul 02841, Republic of Korea; 3Center for Creative Convergence Education, Hanyang University, Seoul 04763, Republic of Korea; hjkhawkes@hanyang.ac.kr; 4Korea BioDefense Research Institute, Korea University, Seoul 02841, Republic of Korea

**Keywords:** kamut sprouts, anti-oxidant, fermentation, cytotoxicity studies

## Abstract

Kamut sprouts (KaS) contain several biologically active compounds. In this study, solid-state fermentation using *Saccharomyces cerevisiae* and *Latilactobacillus sakei* was used to ferment KaS (fKaS-ex) for 6 days. The fKaS-ex showed a 26.3 mg/g dried weight (dw) and 46.88 mg/g dw of polyphenol and the β-glucan contents, respectively. In the Raw264.7 and HaCaT cell lines, the non-fermented KaS (nfKaS-ex) decreased cell viability from 85.3% to 62.1% at concentrations of 0.63 and 2.5 mg/mL, respectively. Similarly, the fKaS-ex decreased cell viability, but showed more than 100% even at 1.25 and 5.0 mg/mL concentrations, respectively. The anti-inflammatory effect of fKaS-ex also increased. At 600 µg/mL, the fKaS-ex exhibited a significantly higher ability to reduce cytotoxicity by suppressing COX-2 and IL-6 mRNA expressions as well as that for IL-1β mRNA. In summary, fKaS-ex exhibited significantly lower cytotoxicity and increased anti-oxidant and anti-inflammatory properties, indicating that fKaS-ex is beneficial for use in food and other industries.

## 1. Introduction

The most common types of wheat are *Triticum durum*, *Triticum spelta*, *Triticum carthlicum*, *Triticum aestivum*, and *Triticum turanicum* (khorasan wheat) [1,2,3]. Among them, khorasan wheat, also known as kamut, is found in food products, and has gained popularity owing to its health benefits, including its anti-inflammatory and anti-oxidant efficacy in kamut cookies, as well as its beneficial effects on diet, compared with modern wheat [4,5,6]. The thousand-year-old khorasan wheat species contains various physiologically active substances such as selenium, β-glucan, GABA, as well as positive health effects [7]. In addition, kamut contains approximately 30% more protein and more fatty acids than regular wheat, and is known for its high nutritional value and ease of digestion [8]. It is an excellent source of proteins, fibers, zinc, phosphorus, magnesium, and vitamins, and is rich in minerals, including selenium [1,7]. These nutrients promote normal cognitive function and metabolism, maintain normal blood pressure, and protect against oxidative stress [9,10,11,12]. However, an excessive intake of kamut can cause side effects, such as vomiting, diarrhea, and hair loss; thus, safety measures need to be considered [13]. Despite these drawbacks, kamut is favored by health-conscious consumers because of its greater health benefits. When the seeds sprout, they biosynthesize essential substances, producing three to four times more vitamins and minerals. In particular, the content of active ingredients in the sprouts that exhibit plant efficacy is often higher than that in adult plants [14]. Thus, sprouted vegetables and plants provide abundant nutrients and are beneficial for human health. Despite the beneficial nutritional and physiologically active substances in sprouts, they can cause abdominal pain and food poisoning [15]. When sprouts germinate, substances that protect and defend themselves against external microorganisms and other external factors are synthesized. They are bioactive substances that are essential for plants. Unfortunately, the higher the component content, the greater the toxic effect. To overcome these obstacles, sprouts are fermented; in fact, those sprout materials are becoming more popular in other industries, especially in cosmetic and medical areas. For example, in fermented *Galactomyces*, ceramide and propanediol are active core ingredients for cosmetic products [16,17,18]. Antibiotics used in the pharmaceutical industry, such as streptomycin, are also produced by fermenting *Streptomyces griseus* [19]. Therefore, various technologies are being applied, with fermentation as a representative example.

Fermentation is one of the oldest techniques in biotechnology, but it is still one of the most actively utilized technologies in new forms, enough to be referred to as fermentology [20]. Currently, the known fermentation technologies include alcohol fermentation and organic acid fermentation, such as lactic acid or acetic acid. One of the most popular fermentation technologies is lactic acid fermentation, which uses lactic acid bacteria to break down sugars. It corresponds to fermented milk products such as cheese and yogurt, and produces lactic acid by the oxidative decomposition of lactose. This fermentation technology contributes to improving fermented foods, including Korean kimchi, soybean paste, natto, jamón, sauerkraut, salami, and tempeh [21]. Ancient Korean pastes and *cheonggukjang* are made through the fermentation of soybeans, resulting in increased vitamin content and other health benefits, such as improved digestive disorders [22,23]. This clearly demonstrates that fermented foods offer more advantages than non-fermented foods [24]; therefore, fermentation technology deserves considerable attention from consumers, companies, and researchers. Since kamut powder is well-used in food materials such as muffins and bread [25], and fermented kamut is also used in wheat dough to make bread in the food industry [26,27], fermenting kamut sprouts may offer different benefits to enhance human health.

Solid-state fermentation (SsF) technology has been applied to maximize the nutritional and physiological properties of kamut. In general, SsF is a simple fermentation method with the advantage that microorganisms are in sufficient contact with oxygen in the air compared to liquid fermentation, and subsequently consume less energy [28,29]. Significant research has been conducted on its efficiency and benefits [30], and many patents have been registered for SsF technology. In the present study, both *Saccharomyces cerevisiae* (*S. cerevisiae*) and *Latilactobacillus sakei* (*La. sakei*) strains were employed in SsF. They have been used as the microbial source for the food fermentation, and their role is to improve the efficacy of fermented food compared to unfermented foods [31,32]. Therefore, KaS and the microorganisms (*S. cerevisiae* and *La. sakei)* were grafted onto SsF technology to observe changes in their anti-oxidant and anti-inflammatory effects. Accordingly, the changes in the efficacy of kamut sprouts before (non-fermented kamut sprout, nfKaS) and after fermentation (fermented kamut sprout, fKaS) were examined in vitro.

## 2. Materials and Methods

### 2.1. Raw Materials and Reagents

Powdered kamut sprout product, used as a raw material, was purchased from Jusung Co. Ltd., Korea. The yeast malt (YM, Lot. 8282839) and Man Rogosa Sharpe (MRS, Lot. 2298709) media that was used for the fermentation were obtained from DIFCO. Food-grade ethanol was used for extraction. The reagents for the β-glucan analysis were obtained by purchasing a β-glucan assay kit (K-YBGL) from Megazyme. Potassium hydroxide and sodium acetate were purchased from DUKSAN, Korea. Folin & Ciocalteu’s phenol reagent (FCR) and gallic acid for the total polyphenol analysis were acquired from Merck, Korea. Sodium carbonate was obtained from Daejung chemicals, Korea. 2,2-Diphenyl-1-picrylhydrazyl (DPPH) and L-ascorbic acid for anti-oxidant efficacy testing were purchased from Merck, Korea. FBS, RPMI-1640 media, as well as Dulbecco’s Modified Eagle’s Medium (DMEM) were purchased from Welgene, Korea, and lipopolysaccharide (LPS), dexamethasone (DEX), and dimethyl sulfoxide (DMSO) were purchased from Merck, Korea. All of the reagents utilized in this research were of extra-pure (analytical) grade.

### 2.2. Microorganisms

*S. cerevisiae and La. sakei* were obtained in the form of frozen vials from the Korean Collection for Type Cultures (KCTC), and *S. cerevisiae* (KCTC 7296) and *La. sakei* (KCTC 3802) were used in the study.

### 2.3. Fermentation of KaS Using Solid-State Fermentation Technology

To determine the optimal conditions during fermentation, the two microorganisms were fermented in three different ways (Table 1). *S. cerevisiae* medium was prepared at 1 × 10^12^ CFU (colony forming units)/mL using YM medium, and *La. sakei* medium was prepared at 1 × 10^12^ CFU/mL using MRS medium. The number of cells was determined by measuring the optical density value (at 600 nm), and also visually confirmed with agar plates. After inoculation, the fermentation was carried out at 30 °C in a fixed state for a total of 6 days. Sampling for the fermenting KaS was performed on the 3rd and 6th days under the three fermentation conditions to obtain the extracts. All of the samples were obtained in 20 g amounts. These three independent conditions were prepared as follows: (1) A volume of 20 mL of YM medium containing 1 × 10^12^ CFU/mL of *S. cerevisiae* was first inoculated to 20 g of dried KaS by spraying at a ratio of 1:1 and fermenting it for 3 days; then, 10 mL of MRS medium containing 1 × 10^12^ CFU/mL of *La. sakei* was further added at a ratio of 1:1:1 (KaS:*S. cerevisiae*:*La. sakei*) by spraying in the same amount ratio compared to the dried KaS used for the first time, and fermented for another 3 days to complete the 6-day fermentation; (2) each medium containing 1 × 10^12^ CFU/mL of both *S. cerevisiae* and *La. sakei* were introduced together to dried KaS at a 1:1:1 ratio (KaS:*S. cerevisiae*:*La. sakei*) by spraying from the beginning and fermenting for 6 days; (3) the same conditions were applied as (1), except the order of the microorganisms. The *La. sakei* was first introduced to dried KaS for 3 days during fermentation, and then *S. cerevisiae* was added and fermented for another 3 days to complete the 6-day fermentation. Upon completion of the 6-day fermentation period, a total of six samples were obtained. Finally, extracts of all of the samples were prepared using the procedure described in Section 2.4.

### 2.4. Preparation of KaS Extracts

After obtaining the fermented KaS samples under three fermentation conditions, 70% (*w*/*v*) ethanol was added to the fermented KaS to concentrate it to 10 times more than the solid content, followed by extracting for 4 h at 25 °C and centrifuging at 3100× *g* to separate the supernatant. The supernatant was evaporated fully using an evaporator (EYELA, Japan), and dried in a vacuum chamber to completely remove solvent for a day. Therefore, KaS extracts fermented after 3 days (pre-fKaS-ex) and all KaS extracts fermented for a total of 6 days were obtained, and non-fermented KaS extract (nfKaS-ex) was obtained via the same extraction process, except for the fermentation process.

### 2.5. β-Glucan

The β-glucan analysis of nfKaS-ex, pre-fKaS-ex, and fKaS-ex was performed by- following the protocol of Lim, C.W, et al. [33]. The amount of sample used for the content analysis of total glucan and α-glucan was 100 mg, and D-glucose solution (1 mg/mL) provided by the assay kit was used as a standard material. The β-glucan analysis is a method in which the total glucan content is analyzed, and then the α-glucan content is subtracted [34]. All tests were performed in triplicate.

### 2.6. Polyphenols

After preparing a 0.1% *w*/*v* gallic acid standard solution in distilled water and diluting it to a concentration of 10 to 100 ppm, 0.6 mL of 2 N FCR was added to a 0.6 g sample, and the mixed solution was reacted at 25 °C for 5 min. Thereafter, a 7% *w*/*v* Na_2_CO_3_ solution and distilled water were added and kept at 25 °C for 90 min [35,36]. After the reaction, the absorbance analyses of nfKaS-ex, preKaS-ex, and fKaS-ex were conducted at 750 nm with a spectrophotometer (Libra S22, Biochrom Ltd., UK). The total polyphenols were calculated and compared with the calibration curve of gallic acid concentration, which was used as the standard [3]. Thus, the units of all results were expressed as mg/g dried weight (dw) of fermented KaS extract. All experiments were conducted in triplicate.

### 2.7. Anti-Oxidant Efficacy Test

To confirm the anti-oxidant effects of the KaS extract before and after fermentation, a DPPH assay was performed. A 1 mM DPPH solution was dissolved in methanol and immediately used, mixed at a 1:1 ratio, and left unexposed to light at 25 °C for 30 min. The absorbance analysis was conducted at 540 nm using a spectrophotometer (Libra S22, Biochrom Ltd., Cambridge, UK). The anti-oxidant efficacy was determined by calculating the radical ratio (%) compared to that of the positive control. The positive control utilized in the test was L-ascorbic acid, and the results were obtained via a comparison analysis for the anti-oxidant effect using the following calculation formula [35]:Free radical scavenging ability (%) = [(blank value − sample value)/blank value] × 100(1)

### 2.8. Cell Cytotoxicity

The Raw264.7 cell-originated *Mus musculus* is a macrophage, and was purchased from the American Type Culture Collection (ATCC), and the HaCaT cell is a human epithelial keratinocyte, and was obtained from the Korea Cell Bank, Korea. Those cells were used to confirm the cytotoxicity of fKaS-ex and nfKaS-ex. fKaS-ex and nfKaS-ex were treated within the Raw264.7 and HaCaT cells, and cell viability was confirmed through a WST-1 assay using the EZ-Cytox cell viability assay kit (Daeil lab service, Seoul, Republic of Korea). Each cell was seeded at 2 × 10^4^ cells/well, and cultured for 1 day at 5% CO_2_ in a 37 °C incubator using a 96-well microplate. After culturing, the medium was exchanged with a serum-free media for starvation, and the cells were cultured for 1 day. After incubation, the treated sample group and control group (untreated) wells were subjected to the addition of EZ-Cytox reagent. The absorbance was measured at a wavelength of 450 nm after 1 h of incubation in a 5% CO_2_ at 37 °C (Multiskan GO, Thermo). Based on the results, the cell viability concentration was determined as a percentage by comparing the cell viability with that of the control group using the formula below [37].
Cell viability (%) = [absorbance of sample added group (450 nm)/absorbance of con-trol group (450 nm)] × 100(2)

### 2.9. Determination of the Anti-Inflammation Effect

To confirm the anti-inflammatory effect, the ability to inhibit nitric oxide (NO) production was confirmed using the Raw264.7 macrophages, and the mRNA expression levels of various cytokines including COX-2, IL-6, as well as IL-1β, were confirmed in the HaCaT cells.

#### 2.9.1. Measurement of NO Values

The Raw264.7 cells were diluted with the fKaS-ex and nfKaS-ex at concentrations 60, 300, and 600 µg/mL using RPMI-1640 medium (contained 1% FBS). Then, 1 µg/mL LPS was administered to all cells, followed by culturing for 1 day on 37 °C (at 5% CO_2_). After incubating with a medium containing NO using Griess reagent, the absorbance was analyzed at a wavelength of 540 nm. Dexamethasone (DEX) served as a positive control. The data were analyzed by calculating the standard curve of the generated NO values.

#### 2.9.2. mRNA Expression Levels of IL-6

The mRNA expression levels of IL-6 in the fKaS-ex and nfKaS-ex groups were determined using a comparative analysis. The HaCaT cells were cultured in DMEM media containing 10% FBS and penicillin/streptomycin (P/S); then, the cells were treated with LPS and incubated at 37 °C for 1 day (at 5% CO_2_).Then, the medium was replaced with serum-free media for starvation followed by each sample dissolved in DMSO then diluted with H2O at different concentration from 600 µg/mL down to 60 µg/ml.. DEX was used as a positive control in the test. Following incubation, the media were washed with PBS. The RNA was isolated from the cells of each sample using the TaKaRa Mini BEST Universal RNA Extraction Kit (Takara, Kusatsu, Japan); then, a Qubit Fluorometer (Thermo Fisher, Waltham, MA, USA) was used for RNA quantification. cDNA was synthesized on an amplifier using 1 µg of each RNA sample. The StepOnePlus™ real-time PCR system (Applied Biosystems, Waltham, MA, USA) was utilized to perform a real-time polymerase chain reaction (RT-PCR). To synthesize the cDNA, a mixed solution of target protein, IL-6 primer, and CyberGreen Power SYBR Green PCR Master Mix (Applied Biosystems, USA) was used. The expression level of IL-6 was measured against GAPDH (glyceraldehyde 3-phosphate dehydrogenase) as an endogenous control for the quantitative RT-PCR analysis [38].

#### 2.9.3. mRNA Expression Levels of COX-2 and IL-1β

The same procedure used to measure IL-6 mRNA was also employed to determine the COX-2 mRNA and IL-1β mRNA expression levels, and the gene expression levels were confirmed using the COX-2 primer and the IL-1β primer instead of the IL-6 primer [38,39]. All of the in vitro tests were executed in triplicate, and the standard deviation (SD) was indicated on the graph.

### 2.10. Statistical Analysis

One-way analysis of variance (ANOVA) was employed to determine differences among the groups after all experiments were conducted in triplicate.

## 3. Results

### 3.1. Optimization of Solid-State Fermentation (SsF)

Three different conditions were tested to determine the optimal fermentation conditions for the KaS. When *S. cerevisiae* and *La. sakei* were inoculated simultaneously at the beginning, and the KaS were fermented for 6 days, the total polyphenol content decreased gradually from 25.2 mg/g dw to 23.4 mg/g dw on day 3, and finally to 22.0 mg/g dw on day 6 (blue triangles in Figure 1). When *La. sakei* was first inoculated and fermented for 3 days, and *S. cerevisiae* was added for further fermentation for 3 days, the total polyphenol content initially decreased from 25.2 mg/g dw to 21.0 mg/g dw on day 3, and further decreased down to 20.8 mg/g dw on day 6 (grey squares in Figure 1). However, when *S. cerevisiae* was first inoculated and fermented for 3 days, and *La. sakei* was added for further fermentation for 3 days, the total polyphenol increased from 25.2 mg/g dw to 25.7 mg/g dw on day 3, and increased again to 26.3 mg/g dw on day 6 (orange circle in Figure 1). Therefore, it is clear that the optimal conditions for SsF technology when fermenting KaS is inoculating *S. cerevisiae* first, fermenting for 3 days, and then introducing *La. sakei* for further fermentation for 3 days. Thus, the following experiments were conducted under the optimized conditions.

### 3.2. Bioactive Substances in Fermented KaS

Since the polyphenol content in fermented KaS increased as fermentation proceeded (Figure 1), the β-glucan contents in all of the nfKaS-ex and fKaS-ex were measured to confirm whether SsF increases bioactive substances of the fermented KaS. The results showed that the concentration of β-glucan was 34.61 mg/g dw when not fermented (nfKaS-ex), 38.45 mg/g dw on day 3 of fermentation (preKaS-ex), and 46.88 mg/g dw after the completion of the 6-day fermentation process (fKaS-ex) (Table 2). Based on these results, the concentration of β-glucan increased by 35.45%, 6 days after fermentation; this is significant because β-glucan is one of the active nutritional contributors to wheat products. β-glucan is also a substance that has a positive effect on improving immune function, especially its inflammatory effects, so the dramatic increase in the amount of the β-glucan content is quite meaningful. These results confirmed that the anti-inflammatory effect of the KaS increased after fermentation. The total polyphenol content in the nfKaS-ex and fKaS-ex was also measured with comparative analysis. The polyphenol content also increased, as the fermentation proceeded similarly as with β-glucan, which was 25.2 mg/g dw nfKaS-ex, 25.7 mg/g dw on day 3 (pre-fKaS-ex), and 26.3 mg/g dw on day 6 (fKaS-ex) (Table 2). Therefore, it clear that the fKaS has positive effects in enhancing bioactive substances.

### 3.3. Anti-Oxidant Efficacy

To determine the anti-oxidant efficacy of KaS, the nfKaS-ex and fKaS-ex were tested using the DPPH assay method over a concentration range of 0.625 mg/mL to 5.0 mg/mL. It was confirmed that both extracts before and after fermentation inhibited free radical generation in a concentration-dependent manner. At 5.0 mg/mL concentrations of nfKaS-ex and fKaS-ex, the free radical scavenging activities were 57.8% *w*/*v* and 71.7% *w*/*v*, respectively (Figure 2). This means that the free radical scavenging ability significantly increased by 24% *w*/*v* after the 6-day fermentation compared to that of the nfKaS. This correlates well with an increase in polyphenols (as observed in Section 3.1), which are widely known as substances with anti-oxidant efficacy. This confirms that as the polyphenol content increases, so does the anti-oxidant efficacy through the SsF of KaS.

Polyphenols are antioxidants that reduce oxidative stress because they contain many hydroxyl groups, which can prevent the formation of reactive oxygen-containing free radicals that destroy cells by attracting the electrons of atoms or molecules bound in cells [40]. Therefore, the higher the polyphenol content, the more enhanced is the anti-oxidant efficacy to protect cells.

### 3.4. Cytotoxicity

Cell cytotoxicity tests were conducted to determine the effect on cell growth of the nfKaS-ex and fKaS-ex in the Raw264.7 and HaCaT cell lines. In the Raw264.7 cells, the cell viability of the nfKaS-ex did not decrease at the lowest concentration, but it started to decrease at a concentration of 0.63 mg/mL, which was 85.3% compared to the untreated group. On the other hand, the cell viability of the fKaS-ex decreased as the concentration increased but still showed more than 100% cell viability (118.6%), even at the greatest concentration (1.25 mg/mL), compared to the untreated group. In general, if the cell viability after the sample treatment was 80% or more compared to the untreated group, it was determined that the extract did not cause cell cytotoxicity. However, if the extract’s concentration was lower, then cytotoxicity existed in the cells. Therefore, these results demonstrated that the nfKaS-ex showed cytotoxicity at a 0.6 mg/mL concentration, but fKaS did not show cytotoxicity even at the 1.25 mg/mL concentration (Figure 3A). In the HaCaT cells, the cell viability of the nfKaS-ex was 83.8% compared to the untreated group at the 2.5 mg/mL concentration. It further decreased to 62.1% at a 5 mg/mL concentration, which demonstrates that cell cytotoxicity began to appear at this concentration. Similarly, with the Raw264.7 cells, the cell viability of the fKaS-ex was still above 80% even at the highest concentration, which was 5.0 mg/mL (88.9%), compared to the untreated group (Figure 3B). Again, these data confirmed that cell cytotoxicity was not observed, even at the highest concentration. As shown in Figure 3A,B, the cell viability of the Raw264.7 cells was much higher (118.6%) than that of the HaCaT cells (88.9%) at the 5.0 mg/mL concentration. However, it is clear that both cell lines depicted cell viability (above 80%), indicating that the fKaS-ex can be an appropriate candidate for food or cosmetic materials because the cell cytotoxicity is very low compared to that of the nfKaS-ex, ensuring the safety of products.

### 3.5. Anti-Inflammatory Efficacy

To confirm the anti-inflammatory effects of the nfKaS-ex and fKaS-ex, an efficacy test was conducted to inhibit NO produced by the Raw264.7 cells (Figure 4). The mRNA levels of COX-2, IL-6, and IL-1β were investigated, due to their significance as essential factors that play a role in regulating the inflammatory response and in cytokines that regulate the immune response (Figure 5A–C).

To measure the NO production, the cells were treated with 1 µg/mL LPS before adding different concentrations of fKaS-ex and nfKaS-ex (60, 300, and 600 µg/mL) for culturing. Then, the NO production levels of the fKaS-ex (60 µg/mL) and nfKaS-ex (600 µg/mL) mixtures were measured to determine whether the NO production was concentration-dependent. For the positive control, 20 µg/mL of DEX was used. As shown in Figure 4, the NO expression level of the nfKaS-ex was 43.3 µM, which was 5.6 µM less than that of the LPS-only treated group. However, the NO expression level of the fKaS-ex was 26.1 µM, which was 22.8 µM lower than that of the LPS-only-treated group at the same concentration. LPS is a lipopolysaccharide that increases cell cytotoxicity; thus, the lower the NO expression, the higher the inhibitory effect. These results indicate that fKas produces less NO than nfKaS, which supports the hypothesis that fermentation is important in lowering NO production, resulting in anti-inflammatory efficacy. To confirm the additional anti-inflammatory effects of the fKaS-ex, the mRNA expression levels of COX-2, IL-6, and IL-1β were evaluated. The cells were exposed to a 1 µg/mL LPS concentration, and then the mRNA expression levels of those cytokines were determined at concentrations of 60 µg/mL to 600 µg/mL of fKaS-ex and nfKaS-ex; the concentration-dependent results were verified. Compared to the untreated group, cells treated with only LPS showed higher cytotoxicity, which was 2.33 times higher than GAPDH (Figure 5A). However, when treated with the 600 µg/mL concentration of nfKaS-ex after LPS treatment, the COX-2 mRNA expression level decreased to 0.82 (35.2% (0.82/2.33 × 100)) compared with the cells with LPS only. When the fKaS-ex was administered at the 600 µg/mL concentration, the COX-2 mRNA expression level compared to GAPDH mRNA decreased to 0.43, which indicates only 18.5% of the COX-2 mRNA was expressed compared with that for the cells treated with LPS only (Figure 5A). These data demonstrate that fKaS lowers the expression of COX-2 mRNA to a greater extent than nfKaS, indicating that fKaS plays an anti-inflammatory role.

To determine the IL-6 and GAPDH mRNA levels, the assay was performed at the same concentrations as those in the COX-2 mRNA analysis; the concentration-dependent results were confirmed. For the nfKaS-ex, the IL-6 mRNA expression level at 600 µg/mL compared to the GAPDH mRNA expression level has decreased to 0.90 from 2.21, showing that 42.5% of the LPS remained in the nfKaS-ex compared with the the original amount (Figure 5B). This indicated that the KaS has an inhibitory effect. However, when the KaS was fermented for 6 days, the IL-6 mRNA expression level of the fKaS-ex showed greater deduction (0.55 from 2.21), showing 26% of LPS remained in the fKaS-ex. This proves that the fKaS-ex has a greater inhibitory effect than that of nfKaS-ex. These results align with the previous COX-2 mRNA level decrease data where the fKaS-ex had a 16.5% higher IL-6 mRNA inhibitory ability compared with the nfKaS-ex.

In the case of IL-1β mRNA, the in vitro test was performed at the same concentrations as the COX-2 and IL-6 mRNA assays, and the results confirmed a concentration-dependent decrease in expression level. In the group treated with a 600 µg/mL concentration of nfKaS-ex after LPS treatment at 1 µg/mL concentration, the IL-1β mRNA expression level slightly decreased to 1.61 compared to the GAPDH mRNA expression level (2.17), in which the LPS concentration was lowered down to 74% compared with the cells treated with LPS only. However, when the KaS was fermented for 6 days, the fKaS-ex resulted in a decrease in the expression of IL-1β mRNA. The expression level of IL-1β mRNA was 0.71 relative to that of GAPDH mRNA (2.17), showing 32.8% of the LPS content compared with the cells treated with LPS only (Figure 5C). These data are the last piece of evidence that clearly confirmed that fKaS-ex can dramatically lower IL-1β mRNA expression compared with nfKaS.

These results aligned and indicated that the expression levels of the cytokines significantly decreased when KaS were added to the cells and fermented for 6 days under all conditions, indicating that the mRNA expression levels of COX-2, IL-6, and 1β all decreased compared to those in cells with nfKaS. The results demonstrated that the anti-inflammatory efficacy significantly improved when KaS was fermented for 6 days. This is not surprising because when Raw264.7 cells are stimulated by LPS, various cytokines such as TNF-α are expressed through cell signaling. In addition, NO, a pro-inflammatory mediator, is also generated. Therefore, if the amount of NO generated is reduced, the inflammatory response is reduced. Similarly to Raw264.7 cells, HaCaT cells also express various cytokines, including IL-6 and IL-1β, due to LPS stimulation. These are widely known inflammatory cytokines, and we confirmed that the expressions of these cytokines were reduced when the cells were treated with fKaS-ex through SsF. Since these are inflammatory cytokines, it can be interpreted that if their expression levels are reduced, the inflammatory response can also be reduced.

## 4. Discussion

Plants and organisms generally have the best growth potential at a young age, and it is widely known that sprouts especially contain more biologically active substances, such as vitamins and minerals, than adult vegetables and plants [14]. This eco-friendly sprouting technique is favored by health-conscious individuals, and kamut is one of the plants used. However, there are toxic substances in sprouts that protect themselves, and their function is to prevent pathogenic microorganism infection from the outside, which can also be harmful to the human body. For example, toxic substances are present in potato sprouts [41]. If the sprouts in potatoes are not removed, health issues such as upset stomach or skin problems may occur. The kamut used in this study is a crop that is known to contain a variety of physiologically active substances, and is known to have anti-oxidant properties and various other effects [1,7]. In addition, KaS also offer various positive effects in humans, but these can also induce negative effects in cells, such as cytotoxicity. Wheat germ agglutinin (WGA) is a protein that is extracted from wheat embryos, and is one of the types of lectins. Lectins are also found in other foods such as beans, bean sprouts, and soybeans [42]. Lectin has a defensive role against pathogenic microorganisms; however, high levels of intake can cause symptoms such as indigestion and diarrhea from irritations of the digestive tract. WGA can bind to cells lining the digestive tract, causing inflammation, and is also associated with respiratory symptoms related to some forms of interstitial lung disease. Additionally, WGA can cause allergic reactions in some people [43]. One key solution to eliminate these negative effects of sprouts is the application of fermentation technologies. Various grains are consumed on a large scale, such as cereals, bread, and other health products, especially in sprout form, because they contain different benefits. However, when sprouted grains are fermented, their health is significantly enhanced. One study showed that when comparing maize and sprouted maize, sprouted maize had more nutritional value. Specifically, when the microbial community was isolated from sprouted maize, microorganisms such as *Latilactobacillus sp*. and *Saccharomyces sp.* were present, but not in whole maize. These microorganisms affect the moisture content, acidity, and sensory properties of the maize. However, when fermented, a higher nutritional value exists. For example, the protein and fat contents are higher in fermented sprouted maize, whereas the fiber and carbohydrate contents are lower in maize. This is because sprouted maize is metabolized through microbial fermentation, resulting in better health benefits in terms of overall acceptability, including color, taste, and flavor [44]. Therefore, this study was conducted to eliminate the drawbacks of sprouts using SsF, one of the most recent fermentation technologies. This study is novel, as it is the first to investigate the properties of sprouted and fermented kamut.

In this study, *S. cerevisiae* and *La. sakei* were selected for KaS fermentation using SsF technology. To measure the effectiveness of fKaS, a comparative analysis of the functional ingredients, cytotoxicity, anti-oxidant, and anti-inflammatory efficacy of fKaS-ex was conducted. *S. cerevisiae* and *La. sakei* are microorganisms that are approved as generally recognized as safe (GRAS), meaning that they are safe for widespread use [45,46]. *S. cerevisiae* is a representative yeast used in food fermentation such as for bread, beer, and wine. It is well known for its ability to break down saccharides and produce alcohol and carbon dioxide during the fermentation process, and it increases the softness of the dough during bread making [31]. *La. sakei* is known to break down proteins during fermentation to synthesize amino acids and other useful substances [32]. Kamut contains about 75% carbohydrates and 18% protein [5], and it was determined that these two microorganisms can effectively utilize compounds in KaS, and have a high potential to increase the efficacy of the fermentation process.

The fermentation patterns differed according to the input method, input time, and the amount of fermentation using *S. cerevisiae* and *La. sakei*. As a result, differences in the pH and fermentation properties may occur depending on the fermentation conditions; these may cause differences in the extraction yield, thus affecting the expected efficacy. Therefore, in this study, the contents of polyphenols and β-glucans, which are bioactive substances, were monitored for 6 days under SsF conditions using *S. cerevisiae* and *La. sakei* in KaS. According to the results (Figure 1), the most suitable fermentation conditions of dried KaS powder are cells fermented using *S. cerevisiae* for 3 days at 30 °C, then introducing *La. sakei* to the cells and fermenting for another 3 days. As fermentation progressed for 6 days, the polyphenol and β-glucan contents also continued to increase. The increase in β-glucan content can also be interpreted as an improvement in extraction efficiency. This is because the large molecular structure of β-glucan is usually broken down into a smaller molecular structure during the fermentation stage, resulting in enhanced bioactive efficacy for anti-inflammatory effects [23]. Research on the results of using more than 6 days of fermentation was not conducted because it has no commercial value in terms of price competitiveness; moreover, contamination or odor issues may occur during long-term fermentation. Thus, in several studies, SsF has been reported to take place for about 2 to 6 days [47,48]. Since 6 days of KaS fermentation was sufficient to remove the negative effects of the sprouts and increase their anti-oxidant effect, it can be expected that fermenting KaS is one of the most promising methods to reap the best benefits of these sprouts.

However, regardless of the effectiveness of a material, safety is an important factor for humans. This means that even if the efficacy is high, if the safety to the human body is low, it is less desirable [49]. Therefore, the question of whether fKaS-ex can be safely improved without the addition of harmful components during the fermentation process remains. Thus, this study conducted a test to investigate whether the cytotoxicity on Raw264.7 and HaCaT cells decreased when the cells were treated with different concentrations of fKaS using SsF technology. The cytotoxicity of the fKaS-ex disappeared at higher concentrations, demonstrating its safety in use in concentrations with decreased cytotoxicity. We confirmed that the expression of the inhibitory effect on various cytokines in the anti-inflammatory effect also increased after fermentation. In general, LPS increases COX-2, IL-6, and IL-1β levels, and induces iNOS expression. This proves that the pro-inflammatory cytokines COX-2, IL-6 and IL-1β were inhibited in their gene expressions when the fKaS-ex was applied.

Various types and pathways of inflammation in humans affect the immune function of the human body. Acute inflammation occurs over a short period, and is typically initiated following injury or infection, while chronic inflammation occurs when repeated inflammatory responses persist. Prolonged inflammation can lead to a decrease in immune function [50]. In this study, anti-inflammatory effects on the skin and the human body can be expected, as fKaS-ex showed a reduced expression level of representative factors of inflammatory cytokines compared to the extract before fermentation. Thus, it can be interpreted that products made of fermented KaS can improve safety while increasing efficacy.

These results are noteworthy, considering the significance of wheat as one of the primary crops, and is also the core ingredient in many foods, both Asian and Western. Therefore, the more benefits wheat offers, the more valuable it is to human health. However, whole wheat contains certain anti-nutritional factors that reduce its nutritional quality, limiting its use in food. Healthier wheat in the food industry can be achieved simply by sprouting and fermenting the grains. In one study, cereal fermentation was performed by inoculating *Latic acid bacteria* [51] on sprouted grains, in order to improve the nutritional value and characteristics; this was shown to result in higher digestibility and lower starch availability. This clearly demonstrates that sprouting and fermentation are safe, inexpensive, and useful technologies for improving functional and nutritional properties [52]. In particular, proteolytic activity increases through the sprouting and fermentation of whole grains, producing bioactive substances such as tyramine, which can be used as a functional food material [53]. Thus, it is not surprising that the sprouted and fermented kamut in this study provided novel insights into reducing cytotoxicity and increasing cell viability by enhancing its anti-oxidant abilities. Furthermore, the advantages of fermentation are well-known in the food and cosmetics industry, leading to a high preference for fermented ingredients. Therefore, fermented KaS material, which significantly reduces cytotoxicity and increases anti-oxidant and anti-inflammatory efficacy, is expected to be a superb material in the food and cosmetics industries.

## 5. Conclusions

Although kamut or kamut sprout has been popular among experts in the food industry, no studies have demonstrated the positive effects of sprouting and fermenting kamut simultaneously. Thus, it is worth highlighting that this novel study demonstrated an increase in anti-oxidant effects, safety, and anti-inflammatory effects by fermenting KaS, especially using SsF technology. In addition, kamut seed and powdered KaS have been actively used in the food industry, and both *S. cerevisiae* and *La. sakei* utilized in this study are registered as GRAS strains, with proven safety in humans. Therefore, it is promising that fermented KaS can be useful for the food industry. Moreover, SsF can be applied to ingredient development in the cosmetics industry because its safety was proven in this study. However, this research was conducted at the cellular level; therefore, clinical trials or in vivo research will be needed for human applications. If successful, this would then open the door to other industries, such as the medical industry, because fermented KaS is food-grade and is suitable for the cosmetics industry.

## Figures and Tables

**Figure 1 foods-12-02107-f001:**
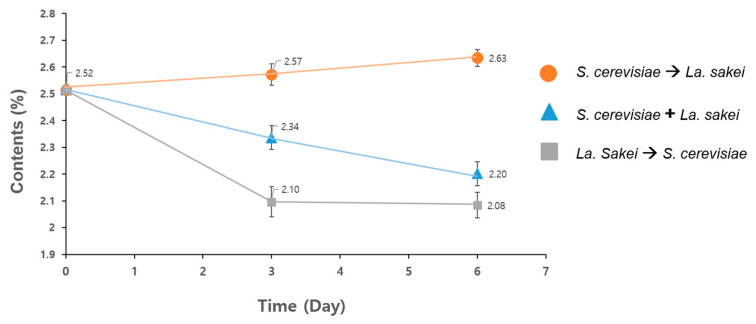
Analysis of total polyphenol content under three fermentation conditions to determine the optimal conditions. Orange circle: first, KaS was fermented by *S. cerevisiae* for 3 days and further fermented by *La. sakei* for 3 days; blue triangle: KaS was fermented together with *S. cerevisiae* and *La. sakei* for 6 days; gray square: first, KaS was fermented by *La. sakei* for 3 days, and then further fermented by *S. cerevisiae* for another 3 days. The one-way ANOVA to confirm significance at a level of *p* < 0.05 was used when comparing the result values among the three different groups. Data are expressed as means ± SD (*n* = 3); significance probability within group was *p* < 0.05.

**Figure 2 foods-12-02107-f002:**
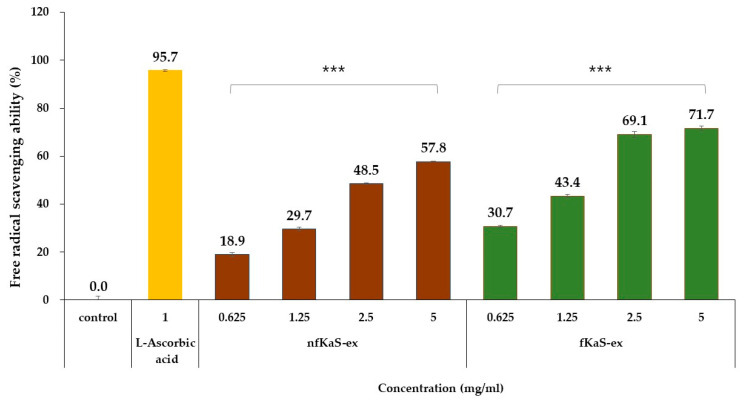
Anti-oxidant effects of nfKaS-ex and fKaS-ex. Control (CTL): no treatment for sample; nfKaS-ex (brown): kamut sprouts extract before fermentation; fKaS-ex (green): fermented kamut sprouts extract for 6 days (*S. cerevisiae* for 3 days and *La. sakei* for 3 days); positive control (yellow): L-ascorbic acid. The free radical scavenging effects of nfKaS-ex and fKaS-ex were compared using a DPPH assay. The data are reported as means ± SD (*p* < 0.001, *n* = 3), and one-way ANOVA analysis was conducted to confirm the differences between groups. Significance probabilities between groups were *** *p* < 0.001.

**Figure 3 foods-12-02107-f003:**
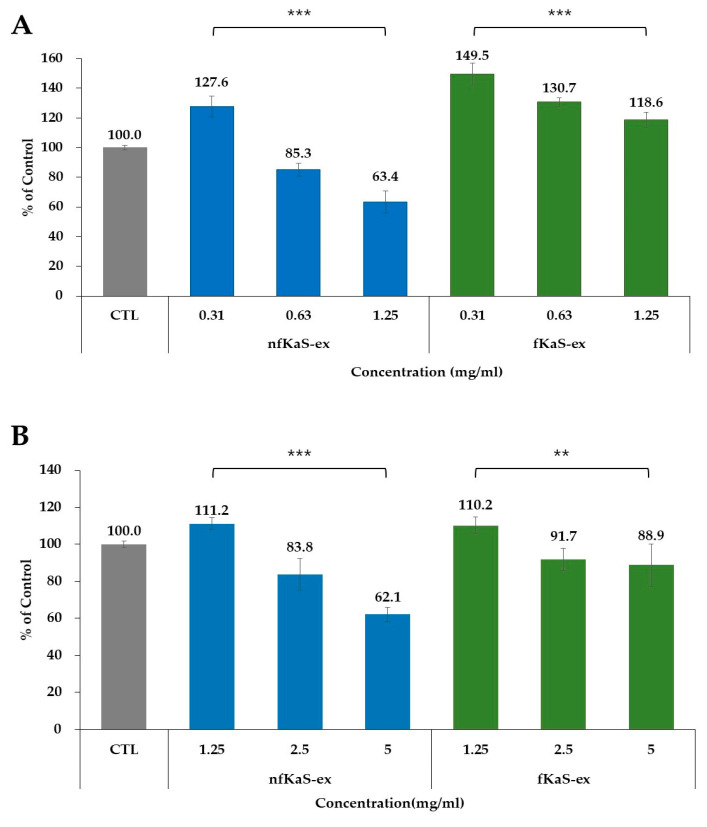
Cytotoxicity results of the nfKaS-ex and fKaS-ex. CTL (gray): no treatment for sample; (**A**) the cytotoxicity results of Raw264.7 cells for nfKaS-ex (blue) and fKaS-ex (green). The fKaS-ex showed higher cell viability. (**B**) The cytotoxicity results of HaCaT cells for nfKaS-ex (blue) and fKaS-ex (green). The fKaS shows cell viability at higher concentrations, similar to the Raw264.7 cytotoxicity results. The data for (**A**,**B**) are reported as means ± SD (n = 3). Significance probabilities between groups (one-way ANOVA): ** *p* < 0.01, *** *p* < 0.001.

**Figure 4 foods-12-02107-f004:**
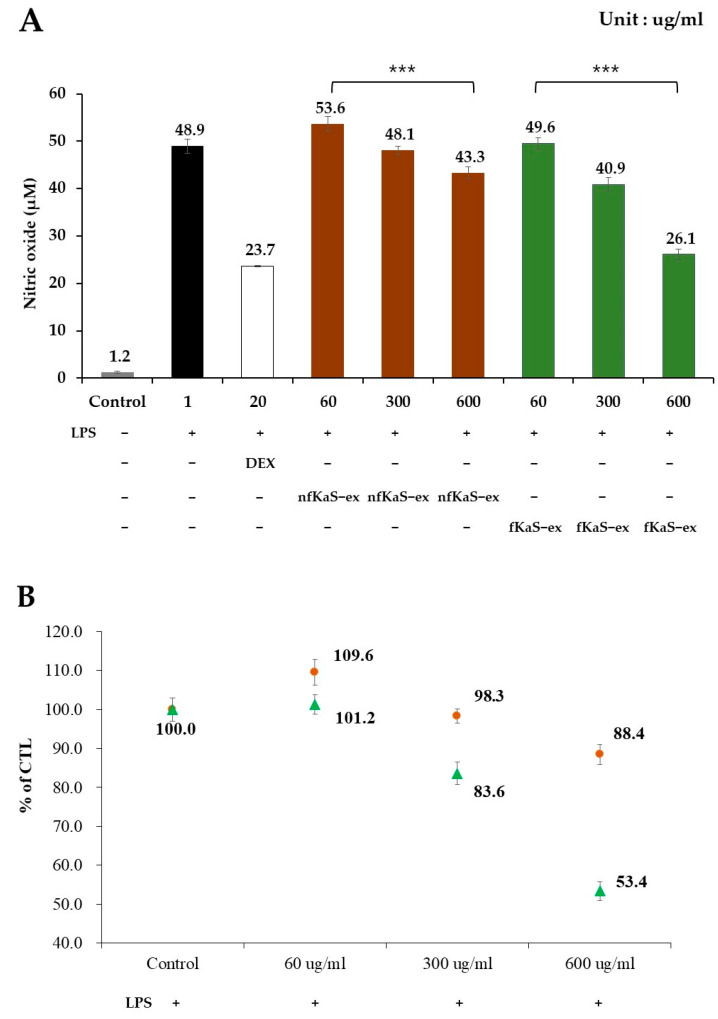
Anti-inflammatory effects of nfKaS-ex and fKaS-ex. (**A**,**B**) Control (gray)—cells were not treated with LPS or the KaS samples. LPS (black)—cells were treated with only LPS. DEX (white)—positive control. nfKaS-ex and fKaS-ex at concentrations of 60, 300, and 600 ug/mL were added to the cells. (**A**) NO assay results for nfKaS-ex (brown) and fKaS-ex (green). After stimulating with LPS for Raw264.7 cells, nfKaS-ex and fKaS-ex were administered to measure NO production. (**B**) Comparison of NO production of the samples treated with nfKaS-ex (brown circle) and fKaS-ex (green triangle) after LPS treatment. Numbers were calculated compared to the control (NO production after LPS treatment in Raw264.7 cells). The mean values along with the SD are reported for all data, and performed in triplicates derived from independent cell cultures. Significance probabilities between groups through one-way ANOVA analysis: *** *p* < 0.001.

**Figure 5 foods-12-02107-f005:**
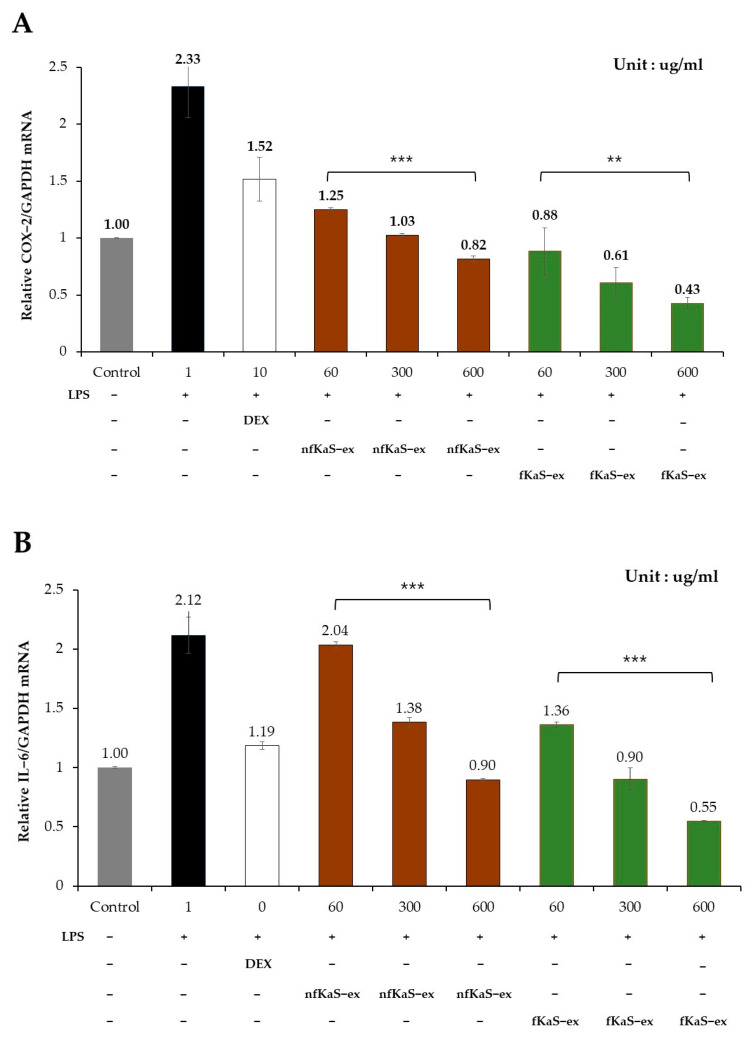
Anti-inflammatory effects of nfKaS-ex and fKaS-ex. (**A**–**C**): Control (gray)—cytokine mRNA expression levels relative to GAPDH mRNA expression levels of cells with no treatment. LPS (black)—only LPS were employed in cells. DEX (white)—positive control. nfKaS-ex and fKaS-ex at concentrations of 60, 300, and 600 ug/mL were added to the cells. (**A**): HaCaT cells were stimulated with LPS, treated with nfKaS-ex (brown) and fKaS-ex (green) to extract cDNA, to confirm the reduction in the COX-2 mRNA expression level compared to the GAPDH mRNA expression level. (**B**): Analysis results of the IL-6 mRNA expression levels for nfKaS-ex (brown) and fKaS-ex (green). (**C**): Analysis results of the IL-1β mRNA expression levels for nfKaS-ex (brown) and fKaS-ex (green). The data are reported as the mean ± SD (n = 3); significance probabilities between groups: ** *p* < 0.01, *** *p* < 0.001 (one-way ANOVA).

**Table 1 foods-12-02107-t001:** Fermentation conditions of dried KaS powder.

Fermentation Condition	Fermentation Time (Day)
Inoculation	1st	2nd	3rd	Inoculation	4th	5th	6th
(1)	*S. cerevisiae*	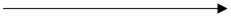	*La. sakei*	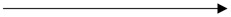
-	-	Sampling	-	-	Sampling
(2)	*S. cerevisiae + La. sakei*	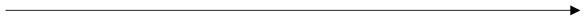
-	-	Sampling	-	-	-	Sampling
(3)	*La. sakei*	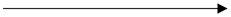	*S. Cerevisiae*	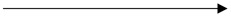
-	-	Sampling	-	-	Sampling

**Table 2 foods-12-02107-t002:** Analysis of β-glucan and total polyphenol content.

Sample	β-Glucan Content (mg/g dw)	Total Polyphenol Content (mg/g dw)
nfKaS-ex ^1^	34.61 ± 0.05	25.2 ± 0.02
pre-fKaS-ex ^2^	38.45 ± 0.04	25.7 ± 0.04
fKaS-ex ^3^	46.88 ± 0.03	26.3 ± 0.03

Data are expressed as means ± SD (*p* < 0.05, *n* = 3). ^1^ nfKaS-ex: kamut sprouts extract before fermentation; ^2^ pre-fKaS-ex: kamut sprout extract fermented for three days using *S. cerevisiae*; ^3^ fKaS-ex: kamut sprouts extract fermented for 6 days (*S. cerevisiae* for 3 days and *La. sakei* for 3 days). The one-way ANOVA analysis was performed to confirm significance for the result comparison among the three different groups.

## Data Availability

The data are contained within the article.

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
