# Peer review of "Fermented Kamut Sprout Extract Decreases Cell Cytotoxicity and Increases the Anti-Oxidant and Anti-Inflammation Effect"

_foods, 2023, doi:10.3390/foods12112107_

Round 1

Reviewer 1 Report

Kamut is found in food products and has gained popularity owing to its greater health benefits, including anti-inflammatory and antioxidant efficacy in kamut cookies, as well as beneficial effects on diet, compared with modern wheat. In this study, the changes in the efficacy of kamut sprouts before (non-fermented kamut sprout, nfKaS) and after fermentation (fermented kamut sprout, fKaS) were examined in vitro. But the fermented kamut sprout contained mixed compounds. So the fermented kamut sprout is not suitable to be used to perform the cell cytotoxicity and anti-inflammation effect.

The comments about the manuscript are as follows: Kamut is found in food products and has gained popularity owing to its greater health benefits, including anti-inflammatory and antioxidant efficacy in kamut cookies, as well as beneficial effects on diet, compared with modern wheat. KaS contain several biologically active compounds. In this study, the changes in the efficacy of kamut sprouts before (non-fermented kamut sprout, nfKaS) and after fermentation (fermented kamut sprout, fKaS) were examined in vitro.

1. Saccharomyces cerevisiae and Lactobacillus sakei were used to ferment KaS by solid state fermentation. Why choose these two strains? The original source of the strains should be presented. 2. The fermentation time is 3 days or 6 days. How is the fermentation time determined? 3. When both microorganisms were introduced. The CFU were presented. But what is the ratio of the Saccharomyces cerevisiae and Lactobacillus sakei? 1:1? 4. The fermented kamut sprout contained mixed compounds. So, the fermented kamut sprout is not suitable to be used to perform the cell cytotoxicity and anti-inflammation effect. It is hard to determine the compounds which increased the antioxidant and anti-inflammatory properties of cell derived from kamut sprout or the inoculated strains. Possibly, the other wheat fermented with Saccharomyces cerevisiae and Lactobacillus sakei have the same properties. So the results and conclusions are unreasonable.

Minor editing of English language required

Author Response

Dear Ms. Danika Zhang ;

May 11, 2023

We are enclosing the revised manuscript entitled “Fermented Kamut Sprout Extract Decreases Cell Cytotoxicity and Increases the Anti-oxidant and Anti-inflammation Effect” along with a separate list of sheets where we have responded the reviewer’s comments as faithfully as we can.

 We have made the following changes in the revised manuscript;

  1. As to the suggestion of the reviewers, we slightly changed some sentences and two native speakers have carefully examined the current manuscript in red color. (Please see the manuscript in red color).
  2. Regarding the plagiarism rate issue, we checked the revised manuscript and found that the plagiarism rate is less than 10%.
  3. Regarding the reviewer’s comments, we showed the statistically significant relationships in the figure or table. All experiments were conducted in triplicate using a statistical program. We showed the statistically significant relationships in the figure or table.

I deeply appreciate your kind consideration of this work and productive comments. We look forward to hearing the positive decision on this revised work. If any questions, please let me know.

Best wishes.

Kwang Yeon Hwang, Professor, Ph.D

Department of Biotechnology, Korea University 02841

Seoul, South Korea

Tel: +82-2-3290-3009

Fax: +82-2-923-3229

==================================================

Reviewer 1

Kamut is found in food products and has gained popularity owing to its greater health benefits, including anti-inflammatory and antioxidant efficacy in kamut cookies, as well as beneficial effects on diet, compared with modern wheat. In this study, the changes in the efficacy of kamut sprouts before (non-fermented kamut sprout, nfKaS) and after fermentation (fermented kamut sprout, fKaS) were examined in vitro. But the fermented kamut sprout contained mixed compounds. So the fermented kamut sprout is not suitable to be used to perform the cell cytotoxicity and anti-inflammation effect.

==> Yes, we agreed. Nevertheless, to overcome these problems and investigate the cytotoxic and anti-inflammatory effects of fermented Kamut sprout for commercial product development, we extracted Kamut sprout and fermentation products under the same conditions using ethanol at the same concentration to study their anti-inflammatory effects and cytotoxicity levels. Therefore, we analyzed samples extracted with the same solvent by solid-state fermentation of Kamut sprout, which we believe is sufficient to compare the cytotoxic and anti-inflammatory effects before and after fermentation.

=====

The comments about the manuscript are as follows: Kamut is found in food products and has gained popularity owing to its greater health benefits, including anti-inflammatory and antioxidant efficacy in kamut cookies, as well as beneficial effects on diet, compared with modern wheat. KaS contains several biologically active compounds. In this study, the changes in the efficacy of kamut sprouts before (non-fermented kamut sprout, nfKaS) and after fermentation (fermented kamut sprout, fKaS) were examined in vitro.

Saccharomyces cerevisiae and Lactobacillus sakei were used to ferment KaS by solid state fermentation. Why choose these two strains?

==> As you may know, Saccharomyces cerevisiae (S. cerevisiae) is a classic yeast used to ferment foods such as bread, beer, and wine. It is well known for its ability to break down carbohydrates (or saccharides) and produce alcohol and carbon dioxide during fermentation, which increases the softness of the dough when making bread. In particular, it is expected to facilitate the extraction of useful substances from the surface of Kamut sprouts through solid-state fermentation. Carbohydrates are the most prevalent component of Kamut, accounting for roughly 75% of its content [1], and are expected to be transformed into monosaccharides by fermentation, which may aid in digestion and absorption as well as promote numerous positive effects [2]. As a result, we undertook a study with S. cerevisiae to evaluate the possibility of increasing these favorable effects.

Furthermore, during fermentation, Lactobacillus sakei (L. sakei) breaks down proteins in meat, creating amino acids and other important chemicals. It also inhibits harmful germs in meat and lowers the pH to securely preserve it [3]. Kamut has roughly 18 % protein [1] which is the second-highest proportion in its overall makeup. We anticipated that the fermentation process would degrade the protein and produce beneficial chemicals, resulting in less inflammation or cytotoxicity. As a result, we conducted a study with L. sakei to look into the potential increase in these beneficial benefits.

Finally, we conducted a study to determine whether synergistic or varied beneficial effects are detected through the fermentation of S. cerevisiae and L. sakei in various ways, both of which are recognized to be helpful bacteria in the human body.

The original source of the strains should be presented.

==> Thank you for your insightful comments. Saccharomyces cerevisiae strains KCTC 7296 and Lactobacillus sakei strain KCTC 3802 were obtained from the Korean Collection for Type Cultures (KCTC). This is indicated in the revision (Lines 102-104, page 3).

The fermentation time is 3 days or 6 days. How is the fermentation time determined?

==> According to the growth curves of S. cerevisiae and L. sakei, they reach the stationary phase 2-3 days following the logarithmic growth phase and die after day 6. As a result, a fermentation time of 6 days was used for this study. While extended fermentation times may be advantageous in terms of research, they may not be economically practical due to market pricing rivalry. As a result, a maximum fermentation time of 6 days was determined for this investigation.

When both microorganisms were introduced. The CFU were presented. But what is the ratio of Saccharomyces cerevisiae and Lactobacillus sakei? 1:1?

==> Yes, we agreed. We inoculated the cultures in a 1:1 ratio of S. cerevisiae and L. sakei.

Regarding as comments, we have supplemented the inoculation ratio (Line 115-120, page 3).

The fermented kamut sprout contained mixed compounds. So, the fermented kamut sprout is not suitable to be used to perform the cell cytotoxicity and anti-inflammation effect. It is hard to determine the compounds which increased the antioxidant and anti-inflammatory properties of cell derived from kamut sprout or the inoculated strains. Possibly, the other wheat fermented with Saccharomyces cerevisiae and Lactobacillus sakei has the same properties. So the results and conclusions are unreasonable.

==> As you mentioned, fermented kamut sprouts are expected to contain a variety of compounds. However, our study did not confirm the anti-inflammatory and cytotoxic effects of fermented Kamut sprouts, but rather compared and analyzed only the soluble substances in ethanol solvent, such as polyphenols extracted from non-fermented kamut sprouts and fermented kamut sprouts. As a result, the results might be interpreted in terms of changes in efficacy caused by fermentation.

Figure 1. The photo of Kamut Seed and Sprout

Additionally, you mentioned, ‘Possibly, the other wheat fermented with Saccharomyces cerevisiae and Lactobacillus sakei have the same properties.’ As shown in Fig.1, the raw material used in this study was Kamut sprouts rather than Kamut seeds. Since the KaS was germinated and not the seeds, it is unlikely that the same efficacy can be attained through the fermentation of wheat seeds alone.

Overall, as mentioned above, we fermented S. cerevisiae and L. sakei, which are recognized as beneficial bacteria for the human body, in various ways to see if we could identify synergistic effects or various beneficial effects.

References

  1. M.H. Abdel-Haleem, A.; A. Seleem, H.; K. Galal, W. Assessment of Kamut® wheat quality. World Rev. Sci. Technol. Sustain. Dev. 2012, 9, 194-203.
  2. Onyema, V.O.; Amadi, O.C.; Moneke, A.N.; Agu, R.C. A Brief Review: Saccharomyces cerevisiae Biodiversity Potential and Promising Cell Factories for Exploitation in Biotechnology and Industry Processes – West African Natural Yeasts Contribution. Food Chemistry Advances 2023, 2, doi:10.1016/j.focha.2022.100162.
  3. Yu, L.; Chen, Y.; Duan, H.; Qiao, N.; Wang, G.; Zhao, J.; Zhai, Q.; Tian, F.; Chen, W. Latilactobacillus sakei: a candidate probiotic with a key role in food fermentations and health promotion. Crit Rev Food Sci Nutr 2022, 1-18, doi:10.1080/10408398.2022.2111402.

Reviewer 2 Report

The present article examines the influence of fermentation on the beneficial properties of sprouts, an interesting topic given the modern consumer's desire to consume wholesome food. I have several comments and questions.

Introduction: lines 56-58: to remove "sweet sugar decomposition, and carbohydrate decomposition", since previously listed are types of fermentations, and listed decompositions occur in the course of the corresponding complex fermentation processes.

Kamut ® is a trademark, so when talking about ancient wheat varieties, better to say khorasan wheat, e.g. on line 31 and line 80. Also, the phrase "ancient microorganisms" is incorrect. Choose whether kamut is lowercase or uppercase and make it consistent in the text

Lactobacillus sakei should be everywhere Latilactobacillus sakei (La. sakei).

You can replace very old references with more recent ones or add more from recent years. Over half of the sources cited are from 10 or 20 years ago, and new information is available, for example, https://onlinelibrary.wiley.com/doi/full/10.1002/jsfa.10402, https://doi.org/10.3390/ foods11243985, https://doi.org/10.3390/foods11233927, https://doi.org/10.3390/nu12041118.

Results: the abscissa and ordinate labels of all figures are very small, please correct them.

Q: Why would you expect the fermented sprouts to show cytotoxicity? Is there any literature on this?

Discussion: Name some of the toxic constituents of wheat germ, as well as literature on the subject.

Line 422: "However, when sprouted grains are fermented, their health is significantly enhanced." : to edit, perhaps referring to health effects?

Line 438-441: the health effect of the species C. cerevisiae and L. sakei cannot be attributed to their "ancient" origin. Please justify the use of these species in some other way, for example, probiotic potential, opportunities to ferment the sugars in the sprouts, or otherwise. These species have undergone no less genetic modification over the centuries than other species of bacteria and yeast, so please do not use unproven claims.

Line 446 - "beta" missing.

Technical error: please add a space before the parenthesis of each reference in the text.

In Conclusion, it is not good to have references and an introduction to the topic again.

The English is readable, but needs polishing.

Author Response

Dear Ms. Danika Zhang ;

May 11, 2023

We are enclosing the revised manuscript entitled “Fermented Kamut Sprout Extract Decreases Cell Cytotoxicity and Increases the Anti-oxidant and Anti-inflammation Effect” along with a separate list of sheets where we have responded the reviewer’s comments as faithfully as we can.

We have made the following changes in the revised manuscript;

  1. As to the suggestion of the reviewers, we slightly changed some sentences and two native speakers have carefully examined the current manuscript in red color. (Please see the manuscript in red color).
  2. Regarding the plagiarism rate issue, we checked the revised manuscript and found that the plagiarism rate is less than 10%.
  3. Regarding the reviewer’s comments, we showed the statistically significant relationships in the figure or table. All experiments were conducted in triplicate using a statistical program. We showed the statistically significant relationships in the figure or table.

I deeply appreciate your kind consideration of this work and productive comments. We look forward to hearing the positive decision on this revised work. If any questions, please let me know.

Best wishes.

Kwang Yeon Hwang, Professor, Ph.D

Department of Biotechnology, Korea University 02841

Seoul, South Korea

Tel: +82-2-3290-3009

Fax: +82-2-923-3229

==================================================

Reviewer 2

The present article examines the influence of fermentation on the beneficial properties of sprouts, an interesting topic given the modern consumer's desire to consume wholesome food. I have several comments and questions.

Introduction: lines 56-58: to remove "sweet sugar decomposition, and carbohydrate decomposition", since previously listed are types of fermentations, and listed decompositions occur in the course of the corresponding complex fermentation processes.

==> Yes, we agreed and applied comments in the revision (Line 56-58 Page 2).

Kamut ® is a trademark, so when talking about ancient wheat varieties, better to say khorasan wheat, e.g. on line 31 and line 80. Also, the phrase "ancient microorganisms" is incorrect. Choose whether kamut is lowercase or uppercase and make it consistent in the text

==> Yes, we agreed and applied comments in the revision. We revised your suggestion, and the case unification of Kamut is now complete.

Lactobacillus sakei should be everywhere Latilactobacillus sakei (La. sakei).

==> Thanks a lot. Yes, we revised your suggestion. We rewrite Lactobacillus sakei as the Latilactobacillus sakei (La. sakei) in the revised manuscript.

You can replace very old references with more recent ones or add more from recent years. Over half of the sources cited are from 10 or 20 years ago, and new information is available, for example,

https://onlinelibrary.wiley.com/doi/full/10.1002/jsfa.10402, / https://doi.org/10.3390/ foods11243985,

https://doi.org/10.3390/foods11233927, / https://doi.org/10.3390/nu12041118.

==> Many thanks. Yes, we revised your suggestion.

Results: the abscissa and ordinate labels of all figures are very small, please correct them.

==>Yes, we revised your suggestion. We redrew all of the figures.

Q: Why would you expect the fermented sprouts to show cytotoxicity? Is there any literature on this?

==> Although it was mentioned during the argument that potatoes are a beneficial crop for people, you are well aware that their sprouts contain a toxic toxin known as solanine [41]. When plants sprout, they have the maximum growth potential and produce a variety of bioactive compounds, including nutrients, for their own growth. During this period, they also produce powerful physiologically active substances to protect themselves from invading diseases, which can be useful but also poisonous to humans. As a result, we looked into whether fermentation could reduce toxicity.

Kimchi and cheese are two examples of food fermentation that increase food shelf life. This implies that including fermentation can improve human body safety. As a result, we undertook research to see if fermented Kamut sprout extract may be used to reduce toxicity.

Discussion: Name some of the toxic constituents of wheat germ, as well as literature on the subject.

==> Yes, we revised your suggestion. Wheat Germ Agglutinin (WGA) is a protein extracted from wheat embryos and is one of the lectins [4]. By irritating the digestive tract, high levels of ingestion can produce symptoms such as indigestion and diarrhea. WGA can bind to digestive system lining cells, causing inflammation, and is also linked to respiratory symptoms associated with some forms of interstitial lung disease. WGA can also induce allergic reactions in certain persons [5].” (Line 431-438, pages 12-13).

References

  1. van Buul, V.J.; Brouns, F.J.P.H. Health effects of wheat lectins: A review. Journal of Cereal Science 2014, 59, 112-117, doi:10.1016/j.jcs.2014.01.010.
  2. Balčiūnaitė-Murzienė, G.; Dzikaras, M. Wheat Germ Agglutinin—From Toxicity to Biomedical Applications. Applied Sciences 2021, 11, doi:10.3390/app11020884.

Line 422: "However, when sprouted grains are fermented, their health is significantly enhanced." : to edit, perhaps referring to health effects?

==> Yes, we rewrote them. As a response to your inquiry, we have expressed that fermentation increases health benefits to the human body from a nutritional standpoint.

Line 438-441: the health effect of the species S. cerevisiae and L. sakei cannot be attributed to their "ancient" origin. Please justify the use of these species in some other way, for example, probiotic potential, opportunities to ferment the sugars in the sprouts, or otherwise.

These species have undergone no less genetic modification over the centuries than other species of bacteria and yeast, so please do not use unproven claims.

==> Yes, as a response to your inquiry, we rewrote them. (Line 458-467, page 13)

“S. cerevisiae and La. sakei are microorganisms that are approved as Generally Recognized as Safe (GRAS), meaning they are safe for widespread use [45,46]. S. cerevisiae is a representative yeast used in food fermentation such as bread, beer, and wine. It is well known for its ability to break down saccharides and produce alcohol and carbon dioxide during the fermentation process as it increases the softness of the dough during bread making [31]. La. sakei is known to break down proteins during fermentation to synthesize amino acids and other useful substances [32]. Kamut contains about 75% carbohydrates and 18% protein [5], and it was determined that these two microorganisms can effectively utilize the compound of KaS and have a high potential to increase efficacy when applied to the fermentation process.”

Line 446 - "beta" missing.

==> Yes, we revised it.

Technical error: please add a space before the parenthesis of each reference in the text.

==> Yes, we revised it.

In Conclusion, it is not good to have references and an introduction to the topic again.

==> Yes, we revised it.

Reviewer 3 Report

This manuscript presents interesting research assumptions related to the biological activity of kamut sprout. However, some contents of the manuscript require correction.

Chapter 2

Materials and methods - no description of the origin of the raw material (wheat). Describe the use of the listed reagents. There is no description of all the reagents used in the study. Please complete.

Chapter 2.2

In what form were the microorganisms purchased

Chapter 2.3

No detailed description of the experiment. Whether the inoculation with microorganisms took place before or after the germination process, or whether the powder was fermented after drying. After sprouting or fermentation, have the sprouts been separated from the grain? How exactly the medium for the microorganisms was prepared, how the microorganisms were introduced whether by spraying or immersion.

Chapter 2.4

To what volume was the supernat concentrated or what procedure was adopted to always evaporate the same volume.

Chapter 2.5

Expand protocol labeled beta glucan

Chapter 2.6

Describe the methods for determining the content of polyphenols, the description provided is incomplete and incomprehensible. Specify the units in which it was expressed.

Chapter 2.7

What spectrophotometer was used

Chapter 2.8

Describe the cell lines used

Chapter 3

Convert w/v units to g or mg sprout dry weight

Chapter 3.3

Notes from Chapter 3

Table 2 convert % units to g or mg

In this section, perform a statistical analysis of the results to provide statistically significant relationships, recommended ANOVA with post-hoc tests.

In the discussion, the authors rely too little on literature data.

Author Response

Dear Ms. Danika Zhang ;

May 11, 2023

We are enclosing the revised manuscript entitled “Fermented Kamut Sprout Extract Decreases Cell Cytotoxicity and Increases the Anti-oxidant and Anti-inflammation Effect” along with a separate list of sheets where we have responded the reviewer’s comments as faithfully as we can.

We have made the following changes in the revised manuscript;

  1. As to the suggestion of the reviewers, we slightly changed some sentences and two native speakers have carefully examined the current manuscript in red color. (Please see the manuscript in red color).
  2. Regarding the plagiarism rate issue, we checked the revised manuscript and found that the plagiarism rate is less than 10%.
  3. Regarding the reviewer’s comments, we showed the statistically significant relationships in the figure or table. All experiments were conducted in triplicate using a statistical program. We showed the statistically significant relationships in the figure or table.

I deeply appreciate your kind consideration of this work and productive comments. We look forward to hearing the positive decision on this revised work. If any questions, please let me know.

Best wishes.

Kwang Yeon Hwang, Professor, Ph.D

Department of Biotechnology, Korea University 02841

Seoul, South Korea

Tel: +82-2-3290-3009

Fax: +82-2-923-3229

======================

Reviewer 3

This manuscript presents interesting research assumptions related to the biological activity of kamut sprout. However, some contents of the manuscript require correction.

Chapter 2

 Materials and methods - no description of the origin of the raw material (wheat). Describe the use of the listed reagents. There is no description of all the reagents used in the study. Please complete.

==> Yes, we revised it. Please check the red colors in the revised manuscript.

Chapter 2.2

In what form were the microorganisms purchased

==> Yes, we revised it. We obtained in the form of freeze vial from Korean Collection for Type Cultures (KCTC). “Powdered kamut sprout product used as raw material was purchased from Jusung Co. Ltd, Korea.” (Line 88 page 2)

Chapter 2.3

No detailed description of the experiment. Whether the inoculation with microorganisms took place before or after the germination process, or whether the powder was fermented after drying. After sprouting or fermentation, have the sprouts been separated from the grain? How exactly the medium for the microorganisms was prepared, how the microorganisms were introduced whether by spraying or immersion.

==>Thanks a lot. Yes, we revised your suggestion. (Please see the red colors in Page 2-4 of revised manuscript.)

Chapter 2.4

To what volume was the supernat concentrated or what procedure was adopted to always evaporate the same volume.

==> Yes, we revised your suggestion. We have supplemented the content according to your advice as follows. “The supernatant was evaporated fully using an evaporator (EYELA, Japan) and dried in a vacuum chamber to completely remove solvent for a day.” (Line 133-134 page 3)

Chapter 2.5

Expand protocol labeled beta glucan

==> Thanks a lot. Yes, we revised your suggestion. (Line 139-144 page 4)

Chapter 2.6

Describe the methods for determining the content of polyphenols, the description provided is incomplete and incomprehensible. Specify the units in which it was expressed.

==> Yes, we revised your suggestion. (Line 146-154 page 4).

Chapter 2.7

What spectrophotometer was used

==> Thanks a lot. Yes, we revised your suggestion. (Line 158-159 page 4). We used a spectrophotometer (Libra S22, Biochrom Ltd., UK)

Chapter 2.8

Describe the cell lines used

==> Yes, we revised your suggestion. (Line 166-179 page 4)

Chapter 3

Convert w/v units to g or mg sprout dry weight

==> Yes, we revised your suggestion. Please see the red colors in revised manuscript.

Chapter 3.3

Notes from Chapter 3, Table 2 convert % units to g or mg

==> Yes, we revised % units to mg units as comments.

In this section, perform a statistical analysis of the results to provide statistically significant relationships, recommended ANOVA with post-hoc tests.

==> Yes, we revised them. Regarding the reviewer’s comments, we showed the statistically significant relationships in the figure or table. All experiments were conducted in triplicate using a statistical program. We showed the statistically significant relationships in the figure or table.

In the discussion, the authors rely too little on literature data.

==> Thanks a lot. Yes, we revised them in the discussion.

Reviewer 4 Report

Main comments: Moderate revision of English style is required.

Methods from 2.3 to 2.7 should be revised: sample preparation procedure and clarification of the point of the method are required.

Abstract

Emphasize the novelty of the study.

Lines 15-16. Unclear expression: what means "..decreased cell viability at 0.63 and 2.5 mg/ml.."? concentration?

Methods

Lines 95-96. Specify commercial No. of used microorganisms.

line 101: unclear statement: the two microorganisms were fermented..

Line 102: what means "introduced"? change the word

Line 106-107: include the commercial no. and producers for YM and MRS media.

lines 121-122. Unclear sentence.

Line 125. Explain the meaning of "extracts were secured"?

Lines 129-132. How the sample for the beta-glucan assay was prepared?

Line 126. Please specify, for what was prepared gallic acid solution, and how the sample for polyphenols analysis was prepared....

Line 142. same comment as above

Lines 175, 183, 200. Please expand the meanings of sub-titles

Results

Line 206. change sub-title to:  Optimization of Fermentation using Solid-state conditions

Fig. 1. please consider a different graphical expression.

Table 2. recommendation to use another expression for the beta-glucan content (g/100 g).

Fig. 2 What about the 'control' sample? specify it

Fig. 3. same comment for the CTL sample.

Moderate revision of English style is required.

Author Response

Dear Ms. Danika Zhang ;

May 11, 2023

We are enclosing the revised manuscript entitled “Fermented Kamut Sprout Extract Decreases Cell Cytotoxicity and Increases the Anti-oxidant and Anti-inflammation Effect” along with a separate list of sheets where we have responded the reviewer’s comments as faithfully as we can.

We have made the following changes in the revised manuscript;

  1. As to the suggestion of the reviewers, we slightly changed some sentences and two native speakers have carefully examined the current manuscript in red color. (Please see the manuscript in red color).
  2. Regarding the plagiarism rate issue, we checked the revised manuscript and found that the plagiarism rate is less than 10%.
  3. Regarding the reviewer’s comments, we showed the statistically significant relationships in the figure or table. All experiments were conducted in triplicate using a statistical program. We showed the statistically significant relationships in the figure or table.

I deeply appreciate your kind consideration of this work and productive comments. We look forward to hearing the positive decision on this revised work. If any questions, please let me know.

Best wishes.

Kwang Yeon Hwang, Professor, Ph.D

Department of Biotechnology, Korea University 02841

Seoul, South Korea

Tel: +82-2-3290-3009

Fax: +82-2-923-3229

=====================

Reviewer 4

Main comments:

The manuscript contains some novelty aspects and can be interesting for readers and researchers.

==> Thank you very much. Your insightful feedback allows us to improve our paper.

Main comments: Moderate revision of English style is required.

==> Yes, we slightly changed some sentences and two native speakers have carefully examined the current manuscript in red color. (Please see the manuscript in red color).

Methods from 2.3 to 2.7 should be revised: sample preparation procedure and clarification of the method description are required.

==>Thanks a lot. We revised them. Regarding as comments, we rewrote the methods (2.3 to 2.7). Please see the red colors in revised manuscript.

Emphasize the novelty of the study.

Lines 15-16. Unclear expression: what means "..decreased cell viability at 0.63 and 2.5 mg/ml.."? concentration?

==>We compare extract concentration to cell viability. The number of healthy cells in a sample is determined using cell viability assays. We made changes based on your valuable comments.

“non-fermented KaS (nfKaS-ex) decreased cell viability from 85.3% to 62.1% at concentrations of 0.63 and 2.5 mg/ml,”

Methods

Lines 95-96. Specify commercial No. of used microorganisms.

==> Yes, we revised it. Saccharomyces cerevisiae strains KCTC 7296 and Lactobacillus sakei strain KCTC 3802 were obtained from the Korean Collection for Type Cultures (KCTC). (Lines 102-104, page 3).

line 101: unclear statement: the two microorganisms were fermented..

==> Thanks a lot. We revised them. Regarding as comments, we rewrote the methods (2.3). Please see the red colors in revised manuscript. (Line 105-126, page 3)

Line 102: what means "introduced"? change the word

==>Yes, we revised it. Regarding as comments, we rewrote the methods (2.3). Please see the red colors in revised manuscript. (Line 105-126 page 3)

Line 106-107: include the commercial no. and producers for YM and MRS media.

==> Thanks a lot. We described in Line 89 of page 2.

lines 121-122. Unclear sentence.

==> Yes, we revised it. We have supplemented the content according to your advice as follows. “The supernatant was evaporated fully using an evaporator (EYELA, Japan) and dried in a vacuum chamber to completely remove solvent for a day.” (Line 133-134, page 3)

Line 125. Explain the meaning of "extracts were secured"?

==> Yes, we revised it. “all KaS extracts fermented for a total of 6 days were obtained”, Line 136, page 3)

Lines 129-132. How the sample for the beta-glucan assay was prepared?

==> Yes, we changed them as your comments. Please see the red colors in revised manuscript. (Line 138-144, page 4)

Line 126. Please specify, for what was prepared gallic acid solution, and how the sample for polyphenols analysis was prepared....

==> We have supplemented about your suggestion. (Line 145-154, page 4)

Line 142. same comment as above

==>We have supplemented about your suggestion. (Line 155-164, page 4)

Lines 175, 183, 200. Please expand the meanings of sub-titles

==> Yes, we changed the sub-titles. (Line 186, 193, 210)

“Measurement of NO values

mRNA expression levels of IL-6

mRNA expression levels of COX-2 and IL-1β”

Results

Line 206. change sub-title to:  Optimization of Fermentation using Solid-state conditions

==> Thanks a lot. We revised to “Optimization of Solid-state Fermentation (SsF)” (Line 219, Page 5)

Fig. 1. please consider a different graphical expression.

==>Yes, we modified it for a clear understanding figure.

Table 2. recommendation to use another expression for the beta-glucan content (g/100 g).

==> Yes, we changed it. We expressed the unit to mg/mg dw (dry weight).

Fig. 2 What about the 'control' sample? specify it

==> Yes, we added it. (Line 287, page 7)

Fig. 3. same comment for the CTL sample.

==> Yes, we added it. (Line 320 page 8)

Reviewer 5 Report

“Fermented Kamut Sprout Extract Decreases Cell Cytotoxicity and Increases the Anti-oxidant and Anti-inflammation Effect” is an interesting and well-structured manuscript. However, several corrections and information must be implemented for the manuscript.

line 23 – keywords – in title, the term anti-oxidant was written with a dash, and in keywords (and text, i.e. line 29) together, please revise the whole manuscript for similar correction issues (the same for sentence lines 72-74)

-line 54 – please put scientific names in italics “Streptomyces griseus” – revise for similar errors

lines 56-58 – the term technologies is used 3 times in this sentence and also fermentation twice. Please rephrase.

line 68 – 70 needs reference and also fermentation also improves the physico-chemical properties of food products: https://www.nature.com/articles/s41598-022-22551-z

line 71 – “Solid-state fermentation” is usually abbreviated as SsF – it would make the manuscript easier to follow

lines 77-78 – “However, in the present study, S. cerevisiae and L. sakei strains were introduced.” – this sentence is not accurate – see references: https://doi.org/10.1016/j.heliyon.2022.e09173, https://doi.org/10.1007/s13205-017-0692-y

lines 78 – 80 – need a reference

line 95 – the manuscript names should be defined at first use in the manuscript and used in shortened way afterwards

line 100 – “Manual of Jusung Co., Ltd” – please specify what kind of instrument this is and also the manufacturer and country of manufacturing (the same for line 147 – spectrophotometer,

line 107 – CFU – please define the abbreviation

line 108 – define MRS

line 109 – prepared or inoculated?

line 111 – please insert a space between the number and degree sign – revise the whole manuscript

line 111 – sampling? for what? define

line 144 – kamut sprout has not been abbreviated? please revise and apply thorough the manuscript

line 204 – please define the number of duplicates or triplicates in case of every experiment and also if statistical analyses were applied – or of not it could be applied to see if there were significant differences between the experiments

line 213 – “decreased initially decreased significantly” – revise and from where do the authors know it decreased significantly if they did no statistical analyses?

figure 1 – revise lines 224-225

figures – please provide higher quality figures

lines 415-417 – is quite similar with lines 31-32 – please revise and don’t duplicate the information

lines 474 – 475 – rephrase

No references should be found in the conclusions section – please revise it and remove all the references or move them to the discussion section!

-          also, in the conclusions appears for the first time that fermented kamut sprouts are food-grade. Please include this information in the introduction with the corresponding reference!

A native English speaker should revise the manuscript as it contains several long and hardly understandable sentences and grammatical errors.

Author Response

Dear Ms. Danika Zhang ;

May 11, 2023

We are enclosing the revised manuscript entitled “Fermented Kamut Sprout Extract Decreases Cell Cytotoxicity and Increases the Anti-oxidant and Anti-inflammation Effect” along with a separate list of sheets where we have responded the reviewer’s comments as faithfully as we can.

We have made the following changes in the revised manuscript;

  1. As to the suggestion of the reviewers, we slightly changed some sentences and two native speakers have carefully examined the current manuscript in red color. (Please see the manuscript in red color).
  2. Regarding the plagiarism rate issue, we checked the revised manuscript and found that the plagiarism rate is less than 10%.
  3. Regarding the reviewer’s comments, we showed the statistically significant relationships in the figure or table. All experiments were conducted in triplicate using a statistical program. We showed the statistically significant relationships in the figure or table.

I deeply appreciate your kind consideration of this work and productive comments. We look forward to hearing the positive decision on this revised work. If any questions, please let me know.

Best wishes.

Kwang Yeon Hwang, Professor, Ph.D

Department of Biotechnology, Korea University 02841

Seoul, South Korea

Tel: +82-2-3290-3009

Fax: +82-2-923-3229

================

 Reviewer 5

“Fermented Kamut Sprout Extract Decreases Cell Cytotoxicity and Increases the Anti-oxidant and Anti-inflammation Effect” is an interesting and well-structured manuscript. However, several corrections and information must be implemented for the manuscript.

line 23 – keywords – in title, the term anti-oxidant was written with a dash, and in keywords (and text, i.e. line 29) together, please revise the whole manuscript for similar correction issues (the same for sentence lines 72-74)

==> Yes, we unified it to ‘anti-oxidant’.

line 54 – please put scientific names in italics “Streptomyces griseus” – revise for similar errors

==> Yes, we changed the microbe names to italics.

lines 56-58 – the term technologies is used 3 times in this sentence and also fermentation twice. Please rephrase.

==> Yes, we rephrased as follows.

“Fermentation is one of the oldest technics in biotechnology, but it is still one of the most actively utilized technologies in a new form, enough to be called as fermentology”

line 68 – 70 needs reference and also fermentation also improves the physico-chemical properties of food products: https://www.nature.com/articles/s41598-022-22551-z

==> Yes, we included the reference.

line 71 – “Solid-state fermentation” is usually abbreviated as SsF – it would make the manuscript easier to follow

==> Yes, we used it to ‘SsF’.

lines 77-78 – “However, in the present study, S. cerevisiae and L. sakei strains were introduced.” – this sentence is not accurate – see references: https://doi.org/10.1016/j.heliyon.2022.e09173, https://doi.org/10.1007/s13205-017-0692-y

==>Yes, we rephrased it. “were introduced à were employed in SsF.”

lines 78 – 80 – need a reference

==> Yes, we rephrased it. “They have been used as the microbial source for food fermentation, and their role is to improve the efficacy of fermented food compared to unfermented foods [31.32]”.

line 95 – the manuscript names should be defined at first use in the manuscript and used in shortened way afterwards

==>  Thanks a lot. Yes, we revised them.

line 100 – “Manual of Jusung Co., Ltd” – please specify what kind of instrument this is and also the manufacturer and country of manufacturing (the same for line 147 – spectrophotometer,

==> Thanks a lot. Yes, we revised them. Please see the red colors in the revised manuscript.

line 107 – CFU – please define the abbreviation

==> Yes, we added it. CFU (Colony Forming Unit) Line 108 page 3.

line 108 – define MRS

==>Thanks a lot. We already described it in Line 89 of page 2.

line 109 – prepared or inoculated?

==> Yes, we revised it. (Inoculating is right).

line 111 – please insert a space between the number and degree sign – revise the whole manuscript

==> Yes, we revised it.

line 111 – sampling? for what? Define

==> Yes, we revised it. “Sampling for fermenting KaS was performed on the 3rd and 6th days under all fermentation conditions to obtain extract.”

line 144 – kamut sprout has not been abbreviated? please revise and apply thorough the manuscript

==> Yes, we revised them.

line 204 – please define the number of duplicates or triplicates in case of every experiment and also if statistical analyses were applied – or of not it could be applied to see if there were significant differences between the experiments

==> Yes, in this study, we conducted triplicates as a basic protocol, and we described them in methods. Please see the red colors in the revised manuscript.

line 213 – “decreased initially decreased significantly” – revise and from where do the authors know it decreased significantly if they did no statistical analyses?

==> Yes, we agreed. We modified the sentence.

figure 1 – revise lines 224-225

==> Yes, we revised the legends.

“Orange circle: first, KaS was fermented by S. cerevisiae for 3 days and further fermented by La. sakei for 3 days; Blue triangle: KaS was fermented together with S. cerevisiae and La. sakei for 6 days; Gray square: first, KaS was fermented by La. sakei for 3 days and then further fermented by S. cerevisiae for another 3 days.”

figures – please provide higher quality figures

==> Yes, we redrew the figures for a clear understanding of them.

lines 415-417 – is quite similar with lines 31-32 – please revise and don’t duplicate the information

==>Yes, we revised them. Thanks a lot.

lines 474 – 475 – rephrase

No references should be found in the conclusions section – please revise it and remove all the references or move them to the discussion section!

==> Thanks a lot. We revised them.

-  also, in the conclusions appears for the first time that fermented kamut sprouts are food-grade. Please include this information in the introduction with the corresponding reference!

==> Yes, we revised them.

Round 2

Reviewer 1 Report

  • The information is more comprehensive. The  study only performed at cell level. It's unreasonable to conclude that SsF can be applied to cosmetics because its safety was proven in this study. I still doubt the extracts of 

    fermented kamut sprout is sitable to perform the cell experiments. Do the S. cerevisiae and La. sakei or their extracts  have the anti-oxidant or anti-inflammation effect?

Author Response

Dear Ms. Danika Zhang ;

May 19, 2023

We are enclosing the second revised manuscript entitled “Fermented Kamut Sprout Extract Decreases Cell Cytotoxicity and Increases the Anti-oxidant and Anti-inflammation Effect” along with a separate list of sheets where we have responded to the reviewer’s comments as faithfully as we can.

I deeply appreciate your kind consideration of this work and productive comments. We look forward to hearing the positive decision on this revised work. If any questions, please let me know.

Best wishes.

Kwang Yeon Hwang, Professor, Ph.D

Department of Biotechnology, Korea University 02841

Seoul, South Korea

Tel: +82-2-3290-3009

Fax: +82-2-923-3229

==================================================

Reviewer 1       

The information is more comprehensive. The study only performed at cell level. It's unreasonable to conclude that SsF can be applied to cosmetics because its safety was proven in this study. I still doubt the extracts of fermented kamut sprout is sitable to perform the cell experiments. Do the S. cerevisiae and La. sakei or their extracts have an anti-oxidant or anti-inflammation effect?

==> Thank you for your insightful comments.  

This study was performed on the cell cytotoxicity, anti-oxidant, and anti-inflammatory effects at the cell level using Kamut sprout extract before and after fermentation, grafting the SsF technique. We agreed with some ambiguous expressions in the conclusion. It is not that the SsF technique is directly applied to cosmetics but rather that it can be employed in the development of cosmetic ingredients. Therefore, we revised the sentence as follows.

Line 537, page 14: “SsF can be applied to ingredient development in the cosmetic industry because its safety was proven in this study.”

As a result of this investigation, we have developed a fermented kamut sprout substance known as "Dr. Kamut.". It is presently promoted as a promising cosmetic ingredient on a global scale.

Do the S. cerevisiae and La. sakei or their extracts have the anti-oxidant or anti-inflammation effect?

==> It has been reported that fruit-derived S. cerevisiae ferment has an anti-oxidant effect [1], and ferment of La. sakei derived from fermented food has also been reported to have anti-oxidant and anti-inflammatory effects [2].

Nevertheless, the study of anti-oxidant and anti-inflammatory effects on the fermentation extracts of S. cerevisiae and La. sakei used in this study was not conducted, as it was not directly aligned with the research topic. Therefore, it is challenging to ascertain the presence of anti-oxidant and anti-inflammatory effects (although it is possible to do so).

However, it was discovered that bioactive components increased inside kamut sprouts by studying changes in total polyphenol and β-glucan content in kamut sprout extracts before and after fermentation. As a result, it is more reasonable to believe that the improved anti-oxidant and anti-inflammatory benefits of kamut sprout were attributable to the increased bioactive substance produced by fermentation, rather than the efficacy of microbial fermentation products themselves.

  1. Fakruddin, M.; Hossain, M.N.; Ahmed, M.M. Antimicrobial and antioxidant activities of Saccharomyces cerevisiae IFST062013, a potential probiotic. BMC Complement Altern Med 2017, 17, 64
  2. Kim, S.; Lee, J.Y.; Jeong, Y.; Kang, C.-H. Antioxidant Activity and Probiotic Properties of Lactic Acid Bacteria. Fermentation 2022, 8,

Reviewer 2 Report

The manuscript is substantially improved. Table 1 should be re-formatted to meet the journal's requirements. Link to all references should be added.

Minor editing of English language required.

Author Response

Dear Ms. Danika Zhang ;

May 19, 2023

We are enclosing the second revised manuscript entitled “Fermented Kamut Sprout Extract Decreases Cell Cytotoxicity and Increases the Anti-oxidant and Anti-inflammation Effect” along with a separate list of sheets where we have responded to the reviewer’s comments as faithfully as we can.

I deeply appreciate your kind consideration of this work and productive comments. We look forward to hearing the positive decision on this revised work. If any questions, please let me know.

Best wishes.

Kwang Yeon Hwang, Professor, Ph.D

Department of Biotechnology, Korea University 02841

Seoul, South Korea

Tel: +82-2-3290-3009

Fax: +82-2-923-3229

==================================================================

Reviewer 2

The manuscript is substantially improved. Table 1 should be re-formatted to meet the journal's requirements. Link to all references should be added.

==> We appreciate your valuable comments. Yes, we revised it.  Line 127, page 3

Reviewer 3 Report

Previous comments have been taken into account. There are no current comments on the manuscript.

Author Response

Dear Ms. Danika Zhang ;

May 19, 2023

We are enclosing the second revised manuscript entitled “Fermented Kamut Sprout Extract Decreases Cell Cytotoxicity and Increases the Anti-oxidant and Anti-inflammation Effect” along with a separate list of sheets where we have responded to the reviewer’s comments as faithfully as we can.

I deeply appreciate your kind consideration of this work and productive comments. We look forward to hearing the positive decision on this revised work. If any questions, please let me know.

Best wishes.

Kwang Yeon Hwang, Professor, Ph.D

Department of Biotechnology, Korea University 02841

Seoul, South Korea

Tel: +82-2-3290-3009

Fax: +82-2-923-3229

==================

Reviewer 3

Previous comments have been taken into account. There are no current comments on the manuscript.

==> We appreciate your valuable comments.

Reviewer 5 Report

The manuscript was considerably improved, the authors implemented all the required corrections, and afterwards, it can be considered for publication.

A small correction, in the conclusion section (Generally Recognized as Safe (GRAS)) it is enough to use the abbreviation only, as it was already abbreviated before

Author Response

Dear Ms. Danika Zhang ;

May 19, 2023

We are enclosing the second revised manuscript entitled “Fermented Kamut Sprout Extract Decreases Cell Cytotoxicity and Increases the Anti-oxidant and Anti-inflammation Effect” along with a separate list of sheets where we have responded to the reviewer’s comments as faithfully as we can.

I deeply appreciate your kind consideration of this work and productive comments. We look forward to hearing the positive decision on this revised work. If any questions, please let me know.

Best wishes.

Kwang Yeon Hwang, Professor, Ph.D

Department of Biotechnology, Korea University 02841

Seoul, South Korea

Tel: +82-2-3290-3009

Fax: +82-2-923-3229

==============

Reviewer 5

The manuscript was considerably improved, the authors implemented all the required corrections, and afterwards, it can be considered for publication.

A small correction, in the conclusion section (Generally Recognized as Safe (GRAS)) it is enough to use the abbreviation only, as it was already abbreviated before

==> We appreciate your valuable comments. We revised it.  Line 535 page 14.